# Transcription-mediated organization of the replication initiation program across large genes sets common fragile sites genome-wide

Olivier Brison[1,2,3,10], Sami El-Hilali [2,3,4,10], Dana Azar [2,3,7], Stéphane Koundrioukoff[1,2,3], Mélanie Schmidt[1], Viola Nähse[2,3,8], Yan Jaszczyszyn[4,5], Anne-Marie Lachages[2,3,9], Bernard Dutrillaux[6], Claude Thermes [4,5], Michelle Debatisse [1,2,3]* & Chun-Long Chen [2,3]*

Common fragile sites (CFSs) are chromosome regions prone to breakage upon replication stress known to drive chromosome rearrangements during oncogenesis. Most CFSs nest in large expressed genes, suggesting that transcription could elicit their instability; however, the underlying mechanisms remain elusive. Genome-wide replication timing analyses here show that stress-induced delayed/under-replication is the hallmark of CFSs. Extensive genome-wide analyses of nascent transcripts, replication origin positioning and fork directionality reveal that 80% of CFSs nest in large transcribed domains poor in initiation events, replicated by long-travelling forks. Forks that travel long in late S phase explains CFS replication features, whereas formation of sequence-dependent fork barriers or head-on transcription–replication conflicts do not. We further show that transcription inhibition during S phase, which suppresses transcription–replication encounters and prevents origin resetting, could not rescue CFS stability. Altogether, our results show that transcription-dependent suppression of initiation events delays replication of large gene bodies, committing them to instability.

[1] CNRS UMR 8200, Gustave Roussy Institute, F-94805 Villejuif, France. [2] Curie Institute, PSL Research University, CNRS UMR 3244, F-75005 Paris, France. [3] Sorbonne University, F-75005 Paris, France. [4] Institute for Integrative Biology of the Cell (I2BC), UMR 9198, CNRS, CEA, Paris-Sud University, F-91198 Gif-sur-Yvette, France. [5] Paris-Saclay University, F-91198 Gif-sur-Yvette, France. [6] CNRS UMR 7205, Museum National d'Histoire Naturelle, F-75005 Paris, France. [7] Present address: Laboratoire Biodiversite et Genomique Fonctionnelle, Faculte des Sciences, Universite Saint-Joseph, 1107 2050 Beirut, Lebanon. [8] Present address: Department of Molecular Cell Biology, Institute for Cancer Research, Oslo University Hospital, Oslo, Norway. [9] Present address: UTCBS, CNRS UMR 8258/ INSERM U 1267, Sorbonne-Paris-Cité University, F-75006 Paris, France. [10] These authors contributed equally: Olivier Brison, Sami El-Hilali. *email: michelle.debatisse@gustaveroussy.fr; chunlong.chen@curie.fr

The entire genome should be duplicated once and only once during each cell cycle to maintain genome integrity. Under normal growth conditions, the firing of tens of thousands of adequately distributed replication origins is needed to ensure the proper duplication of the human genome before mitotic onset. In cells exposed to mild replication stress, the overall replication rate is supported by the recruitment of a large pool of extra-origins, an adaptation process called compensation[1]. Nevertheless, conventional cytogenetic analyses have shown that in stressed cells, some regions of the genome, notably common fragile sites (CFSs), display breaks in metaphase chromosomes, suggesting that replication is recurrently not completed at these regions[2]. Therefore, CFSs are major drivers of genome instability and subsequent chromosome alterations associated with human diseases, notably cancer[3]. CFSs have long been associated with very large, mega-base (Mb)-sized genes[4], some of which behave as tumour suppressors[5,6]. In addition, many very large genes have been associated with inherited diseases, such as neurodevelopmental and neuropsychiatric disorders[7].

Molecular mapping of CFSs in different human cell types has further extended the correlation between large genes and CFSs to genes more than 300 kb long and, importantly, pointed out that CFSs display tissue-specific instability[8]. Strikingly, these genes and the associated CFSs are conserved in mouse[4,8] and chicken cells[8,9]. A pioneer work focussing on five large genes has shown that CFSs are instable only in cells where the corresponding genes are expressed[10]. A recent chromatin immunoprecipitation-sequencing (ChIP-Seq) analysis of FANCD2, a factor that binds preferentially to CFSs from S-phase to mitosis upon replication stress[11], has confirmed genomewide that CFSs colocalize with large transcribed genes in human[12,13] and in chicken cells[9] grown in vitro. Remarkably, this correlation between CFSs and large transcribed genes has been detected also by extensive mapping of copy number variations in a large series of tumours[14]. Therefore, it is now clear that transcription plays a major role in CFS setting, thus explaining why different subsets of large genes are fragile in different cell types.

Two main models have been proposed to explain the susceptibility of large expressed genes to replication stress. The first one relies on the hypothesis that transcription of such genes takes more than one cell cycle, and that, consequently, the transcription and replication machineries will necessarily encounter during S phase[10]. When replication and transcription interfere, the most deleterious situation arises upon head-on encounters that might favour the formation and stabilization of R-loops, namely DNA/RNA hybrids resulting from annealing of the nascent transcript with the template DNA strand[15,16]. It was therefore proposed that R-loops frequently form in the body of large genes, which delays fork progression and leads to CFS under-replication upon replication stress[17]. A variant of this model proposes that stress-induced uncoupling of the replicative helicase from DNA polymerases gives rise to single-stranded DNA, which elicits the formation of DNA secondary structures at particular sequences, notably at stretches of AT dinucleotides. These structures would further delay polymerase progression, increasing the frequency of replication–transcription encounters[18,19].

The second model is based on molecular combing analyses of the distribution of initiation events along three CFSs, showing that the body of the hosting genes is origin-poor under normal growth conditions and/or upon stress. Consequently, these genes are replicated by long-travelling forks emanating from the flanking regions[11]. The strong delay of replication completion that specifically occurs upon fork slowdown might elicit CFS instability. In this scenario, the role of transcription could be to remove origins from the gene body, as previously shown in various models[20–24]. A direct support to this view was recently provided experimentally by replacing the endogenous promoter of three large genes with promoters of various strengths, which has shown that the transcription level dictates the density of initiation events across the gene body[25]. However, many large transcribed genes escape fragility, indicating that transcription per se is not sufficient to commit them to instability[8].

Replication timing is another parameter often evoked to explain CFS susceptibility to replication stress. Indeed, the risk of being under-replicated at mitotic entry might be higher for late-replicating regions[11]. Nevertheless, it is not clear whether and how the replication timing affects CFS stability. Here we report Repli-Seq analysis of the replication programme in human lymphoblasts grown in the absence or in the presence of aphidicolin (Aph), an inhibitor of replicative DNA polymerases used in vitro to destabilize CFSs[2]. We identified regions that displayed specific replication delay upon Aph treatment, resulting in under-replication. More than 80% of these delayed/under-replicated regions nest in chromosome domains that are transcribed continuously across at least 300 kb and poor in replication initiation events, and thus are replicated by long-travelling forks. Strikingly, the orientation of these forks relative to transcription is neutral for establishment of delayed/under-replication. We further showed that these regions correspond to a major class of CFSs. Noteworthy, inhibition of transcription in cells already engaged in the S phase, a condition that suppresses transcription–replication encounters and prevents origin resetting[26], did not alleviate CFS instability. Altogether, our results demonstrate genome-wide that transcription-dependent segregation of initiation events out of the gene body generates long-travelling forks in large transcribed domains, which elicits the replication timing delay responsible for CFS instability upon replication stress.

## Results

**Impact of replication stress on the replication dynamics**. Using the Repli-Seq technique (Fig. 1a and Supplementary Fig. 1a), we first determined the replication timing profile of human lymphoblastoid JEFF cells grown under normal conditions (Methods). The profiles were highly reproducible between three biological replicates (Pearson's $R > 0.97$, $P < 10^{-15}$) (Supplementary Fig. 1b, NT1-3) and very similar to those previously reported for GM06990 and GM12878 cells, two other lymphoblastoid cell lines[27] ($R > 0.93$, $P < 10^{-15}$) (Supplementary Fig. 1b). These data therefore confirm that Repli-Seq is a robust technique, and that the replication timing programme is well conserved between different isolates of the same tissue[28]. We then analysed two biological replicates ($R = 0.95$, $P < 10^{-15}$) (Supplementary Fig. 1b, Aph1-2) of JEFF cells treated with 600 nM of Aph for a total of 16 h prior to cell sorting (Fig. 1a), a stress condition commonly used to induce breaks at CFSs. To determine the effect of Aph-induced stress on replication dynamics, we computed the mean number of reads per 50 kb window for control cells (NT1-3) and the Aph-treated cells (Aph1-2), and calculated an under-replication index (URI) defined as the Z-score of the difference between the sum of reads per window in cells treated (Aph) or not (NT) with Aph ($\Delta_{\text{Aph-NT}}$, Methods). Negative URI ($\Delta_{\text{Aph-NT}}$) identified regions delayed/under-replicated upon stress.

**Fork slowing strongly affects the timing of specific loci**. We then compared the URI with the replication timing, calculated as the S50 (on a scale from 0, early, to 1, late), which is the moment in S phase when a sequence has been replicated in 50% of cells (Methods). We found that approximately half of the genome, essentially domains with $S50 \geq 0.5$ (mid-late and late), displayed $\text{URI} \leq 0$, whereas the rest of the genome showed $\text{URI} \geq 0$ (Fig. 1b). Importantly, we identified 330 highly delayed/under-

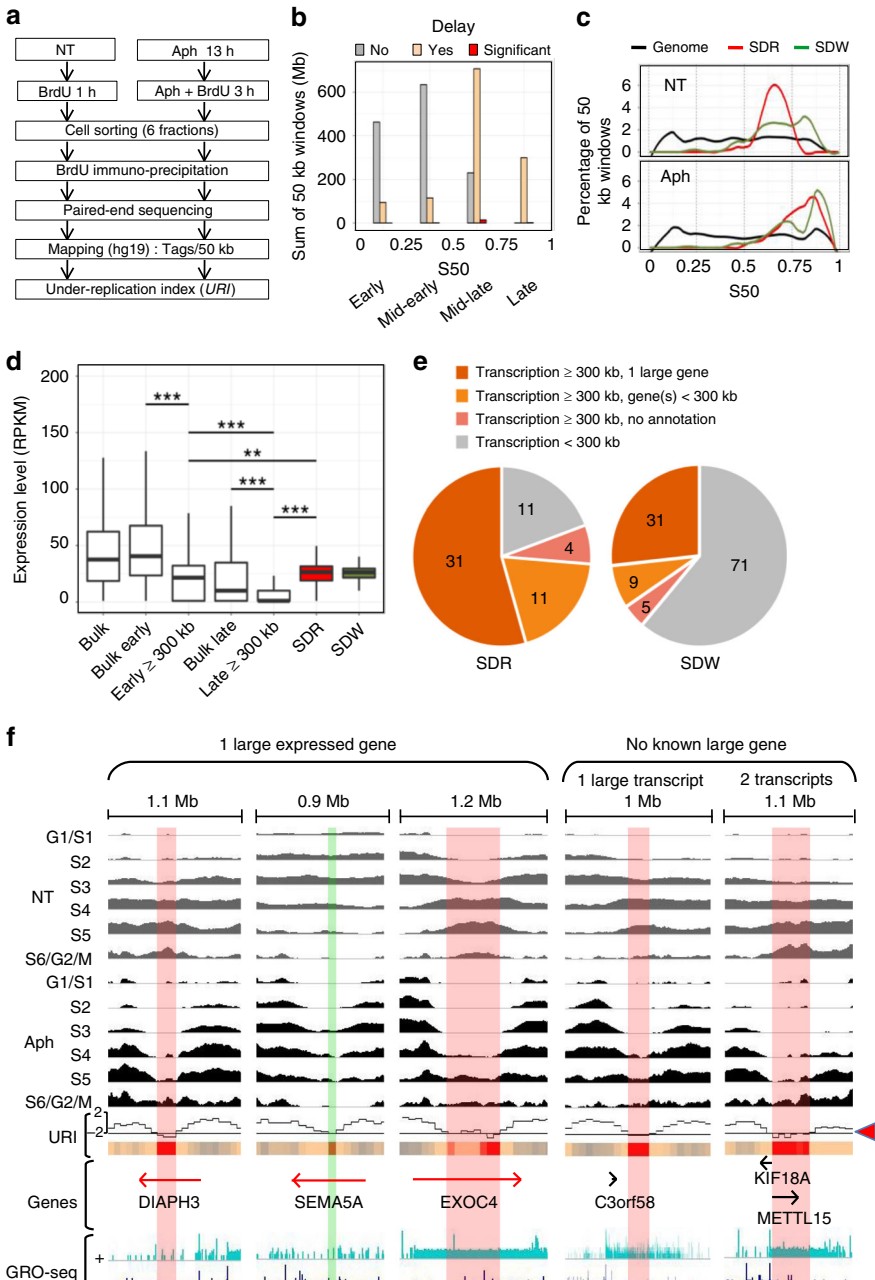

**Fig. 1 Genomewide profiling of replication timing identifies SDRs/SDWs. a** Scheme of Repli-Seq experiments with untreated (NT) and aphidicolin-treated (Aph) lymphoblasts. The Under-Replication Index (URI) was calculated at 50 kb resolution ($\Delta_{Aph-NT}$, Methods). **b** Histogram showing the sum of 50 kb windows (in Mb) displaying no delay (URI > 0, $n = 26{,}558$, 1328 Mb), low-to-moderate delay ($-2 < URI < 0$, $n = 24{,}343$, 1217 Mb), or significant delay (URI < −2, $P < 0.05$, $n = 330$, 16.5 Mb) upon Aph treatment in four timing classes (defined by the S50 on a scale from 0 to 1). **c** Percentage of 50 kb windows in SDRs ($n = 57$), isolated SDWs ($n = 116$), and along the genome ($n = 56{,}783$) as a function of S50. These percentages were computed within each bin (bin size = 0.01) then Loess smoothed ($\alpha = 0.25$). For SDRs, the values correspond to the mean timing of all 50 kb windows in each SDR. **d** Boxplot (bounds of box: 25th and 75th percentiles; centre line: median) showing the gene expression level distribution relative to their replication timing. For genes > 50 kb, the timing values correspond to the mean timing of all 50 kb windows enclosed in each gene. The expression levels (RPKM, Methods) were measured by GRO-Seq[29] for all expressed genes (Bulk, $n = 15{,}204$), all early-replicating (S50 < 0.5, $n = 13{,}257$), all late-replicating (S50 ≥ 0.5, $n = 1947$), the subset of early-replicating ($n = 315$) or late-replicating ($n = 304$) genes ≥ 300 kb free of SDR/SDW, and for genes harbouring an SDR ($n = 54$) or an SDW ($n = 39$) (Kolmogorov–Smirnov test: **$P < 0.01$; ***$P < 0.001$). **e** Pie-chart showing the numbers of SDRs or SDWs associated with the indicated classes of transcripts. **f** Representative examples of SDRs/SDWs nested in large transcribed domains. The Repli-Seq profiles (NT and Aph) were normalized to the same background level (Methods). The URI profile is shown with red arrowhead pointing to −2 threshold (Methods), aligned with the corresponding heat-map (colours as in **b**). Genes and the GRO-Seq profiles on the Watson (+) and Crick (−) strands are shown. The SDRs (red vertical bars) and SDW (green vertical bar) are indicated. The genomic regions displayed are from left to right: chr13:61-61.1; chr5:8.8-9.7; chr7:132.8-134; chr3:143.4-144.4; chr11:27.7-28.9 Mb. Source data are provided as a Source Data file.

replicated windows (URI ≤ −2, $P < 0.05$), called below significantly delayed windows (SDWs), among which 314 reside in domains with a S50 ≥ 0.5 (Fig. 1b). Genome-wide clustering of SDWs identified 57 regions containing at least two SDWs separated by <250 kb (see Methods for the choice of this threshold), a distance significantly smaller than what expected for random distribution (Kolmogorov–Smirnov (KS) test $P < 10^{-15}$) (Supplementary Fig. 1c, left panel). These regions, named significantly delayed regions (SDRs), may include up to 16 SDWs and may extend over hundreds of kb (Supplementary Fig. 1c right panel and Supplementary Data 1). Overall, SDRs enclose 214 SDWs and 116 SDWs remained isolated (Supplementary Data 2).

The timing profile of untreated cells shows that isolated SDWs are equally distributed between mid-late ($0.5 ≤ S50 ≤ 0.75$, $n = 62$) and late ($0.75 ≤ S50 ≤ 1$, $n = 40$) replicating domains and are further delayed upon stress (Fig. 1c and Supplementary Data 2). By contrast, 56 out of 57 SDRs (98%) nest in mid-late domains in untreated cells (Fig. 1c and Supplementary Data 1). Upon stress, the SDRs however present S50 values resembling those of late-replicating domains (Fig. 1c), showing that they are much more delayed than all other mid-late regions. Similar results were obtained with three different methods, indicating that they cannot be explained by a normalization bias (Supplementary Fig. 1d).

**SDRs/SDWs nest in large transcribed late-replicating domains**. We then looked for potential correlations between SDRs/SDWs and transcription features. Analysis of data obtained by GRO-Seq (Global Run-On Sequencing, a method to measure nascent RNA) from untreated GM06990 lymphoblasts[29] showed that 31 SDRs nested within large ( > 300 kb) expressed genes (Figs. 1d–f). The human genome contains ~890 annotated genes larger than 300 kb, among which about 57% display an S50 ≥ 0.5, including those hosting SDRs. We observed that large genes, whatever their replication timing, were modestly transcribed compared with the bulk genes, and that early-replicating large genes were more transcribed than late-replicating ones (Fig. 1d). Noticeably, the expression level of genes hosting an SDR and/or an SDW is similar to that of early-replicating large genes, and thus significantly higher than that of other late-replicating large genes ($P < 10^{-5}$) (Fig. 1d).

We also found that four SDRs nested within large regions displaying strong GRO-Seq signal coming either from still non-annotated genes or from non-coding sequences and 11 SDRs were within long-transcribed domains (>300 kb) harbouring 2–3 adjacent genes, the individual size of which could be smaller than 300 kb (Fig. 1e, f and Supplementary Data 1). Together, 46 out of 57 SDRs (81%), named below transcription-associated SDRs (T-SDRs), nest in chromosome domains transcribed across at least 300 kb. Although the correlation appears less striking, 45 isolated SDWs (39%), named below T-SDWs, also nest in transcribed domains > 300 kb, which show similar features as the SDR-hosting domains (Fig. 1e, f and Supplementary Data 2). The proportion of isolated SDWs not associated with large genes is therefore significantly higher than that of SDRs, suggesting that SDWs mark heterogeneous categories of delayed sequences. We focused below on the 45 T-SDWs, among which 30 display at least one nearby window with an URI close to −2. The latter T-SDWs could therefore be false-negative T-SDRs, resulting from our stringent −2 URI cut-off (see *SEMA5* in Fig. 1f). In conclusion, T-SDRs and T-SDWs (T-SDRs/SDWs) thus extend in moderately expressed large genes/domains, the body of which replicates in the second half of S phase in normal conditions and displays strong delayed/under-replication upon stress. Conversely, transcribed large genes, the replication of which is completed

before S6/G2/M upon stress, and non-transcribed large genes, even late replicating, do not show under-replication (Supplementary Fig. 1e).

**T-SDRs/SDWs nest in domains poor in initiation events**. We then analysed replication initiation in T-SDRs/SDWs and their flanking regions using data available for untreated GM06990 lymphoblasts. Analysis of Bubble-Seq data[30] showed that over 80% of T-SDRs/SDWs, as well as their surrounding regions (several hundreds of kb to >1 Mb), were poor in initiation events when compared with the genome-wide distribution (KS test $P < 10^{-15}$) (Fig. 2a). This finding was further confirmed and extended by analysis of replication fork directionality (RFD) (Fig. 2b) determined by Okazaki fragment sequencing (OK-Seq)[31]. In most cases, we observed that two major initiation zones flank the large transcribed genes hosting T-SDRs/SDWs. In general, one of these initiation zones overlaps with the gene promoter, whereas the second one lies at variable distance from the gene 3′-end (Figs. 2c and 3a). Because of the large gene size, unidirectional forks emanating from these zones travel across several hundreds of kb to complete replication of the gene body.

Although the body of T-SDR/SDW-hosting genes replicates in the second half of S phase in normal growth conditions, we observed that their flanking initiation zones often fired early (NT in Figs. 1f, 2c and 3a), sometimes very early in the S phase (Fig. 2c, *FHIT*). A previous analysis of unstressed cells by molecular combing has shown that the *FHIT* gene displays an initiation poor core extending for about 800 kb, and that replication forks travel along the gene at ≈1.8 kb/min, like in the bulk genome[11]. In these conditions, convergent forks would need 8–9 h to complete *FHIT* replication, in agreement with the replication kinetics observed here (NT in Fig. 2c). Therefore, in addition to the firing time of the initiation zones flanking this large gene, the distance that convergent forks must travel before merging strongly contributes to set the replication timing of the gene body in untreated cells. We found here that this feature is common to large expressed genes (NT in Figs. 1f, 2c and 3a). Often, replication could not be completed when fork speed is reduced upon treatment with Aph (Aph in Fig. 1f, 2c and 3a), which gives rise to the T-SDRs/SDWs. The distance separating the initiation zones flanking the genes is therefore a major parameter for T-SDRs/SDWs setting. It is noteworthy that although poor in initiation events, the body of T-SDR/SDW-hosting genes could display weak initiation zones firing from S4 to S6. These initiation events tend to increase the URI locally and therefore help replication to proceed across large genes (Fig. 1f, 2c and 3a). We conclude that initiation paucity and subsequent long-travelling forks are causal to T-SDR/SDW under-replication.

**T-SDR localization depends on the flanking initiation zones**. The OK-Seq profiles show that the T-SDRs/SDWs may lie at the centre of the large fragile genes or in an asymmetric position (Fig. 2c and Supplementary Figs. 2a and 3a). Not surprisingly, comparison of the Repli-Seq and OK-Seq data shows that centred T-SDRs/SDWs correlate with convergent forks travelling similar distances in the genes before merging in untreated cells (Fig. 2c left panel and Fig. 3a), whereas T-SDRs/SDWs are asymmetric when convergent forks travel different distances. In the latter cases, the T-SDRs/SDWs are most often positioned close to the 3′-end of the gene, because the 5′-initiation zone generally fires first and more efficiently than the 3′-one. In these cases, replication forks that travel the longest distances emanate from the gene promoter and progress co-directionally with transcription (Fig. 2c right panel and Fig. 3a). The opposite situation was observed in only two cases (Supplementary Fig. 2a). Together, our

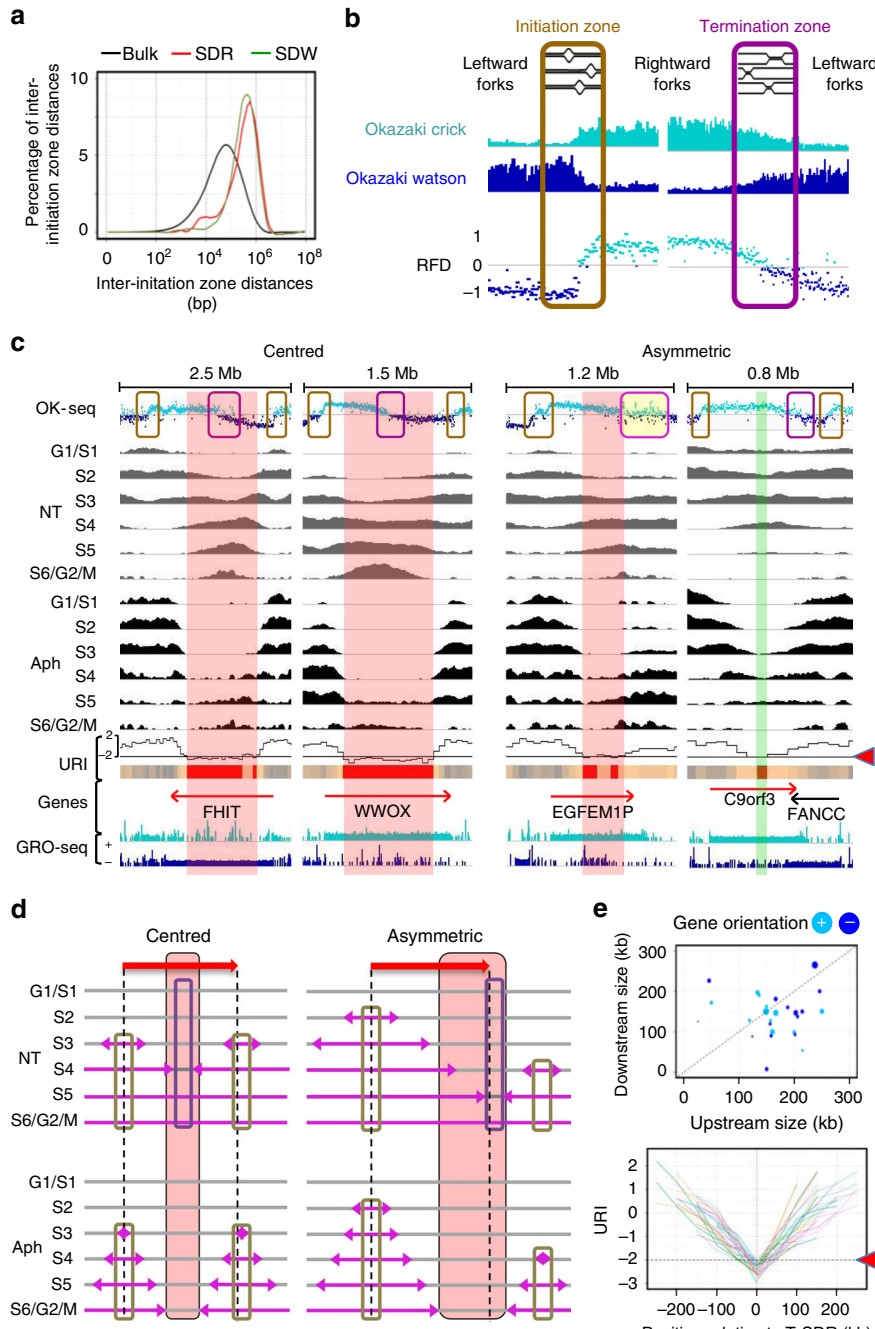

**Fig. 2 T-SDR/SDW localization relies on the properties of flanking initiation zones. a** T-SDRs/SDWs are poor in initiation events. Percentage of inter-initiation zone distances identified by Bubble-Seq[30] for the bulk genome and regions overlapping SDRs or SDWs. The percentages were computed as in Fig. 1c with bin size = 0.1. **b** Determination of replication fork directionality (RFD) by Okazaki fragment sequencing (Ok-Seq). Each point shows the RFD value, computed (in 1 kb windows) as the difference between the proportions of forks moving rightward (fragments mapping on the Crick strand) or leftward (fragments mapping on the Watson strand). Upward (downward) transitions correspond to initiation (termination) zones[31]. **c** Representative examples of genes ≥ 300 kb (red arrows) displaying a centred (left panels) or an asymmetric (right panels) termination zone. The initiation and termination zones are highlighted by orange and purple boxes on the OK-Seq profiles, as in **b**, the yellow-filled box corresponds to a large zone showing complex RFD pattern with both initiation and termination events. Repli-Seq, URI and GRO-Seq profiles, and T-SDRs/SDWs, are as in Fig. 1f. From left to right, regions are chr3:59-61.5; chr16:77.9-79.5; chr3:167.6-168.9; chr9:97.4-98.2 Mb. **d** Scheme depicting how initiation poor regions nested in a large gene body elicit under-replication upon fork slowing. Pink arrows: replication forks. Red arrows and vertical dotted lines: genes. Initiation and termination zones, and the T-SDRs/SDWs are represented as above. Left: initiation zones with comparable firing times and efficiencies flank the large gene. Right: the flanking initiation zones display different firing time and localization relative to the 5′- and 3′-ends of the gene. **e** Upper panel: dot-plot showing the distances separating the transcription start site from the upstream end (Upstream size) and the transcription termination site from the downstream end (Downstream size) of the T-SDR (n = 46). The circle sizes are proportional to the gene lengths. Genes on the + or – strand are indicated (light and dark blue, respectively). Lower panel: each colour line shows the URI profile of the upstream and downstream (negative and positive values on the x axis, respectively) regions flanking each individual T-SDR. Source data are provided as a Source Data file.

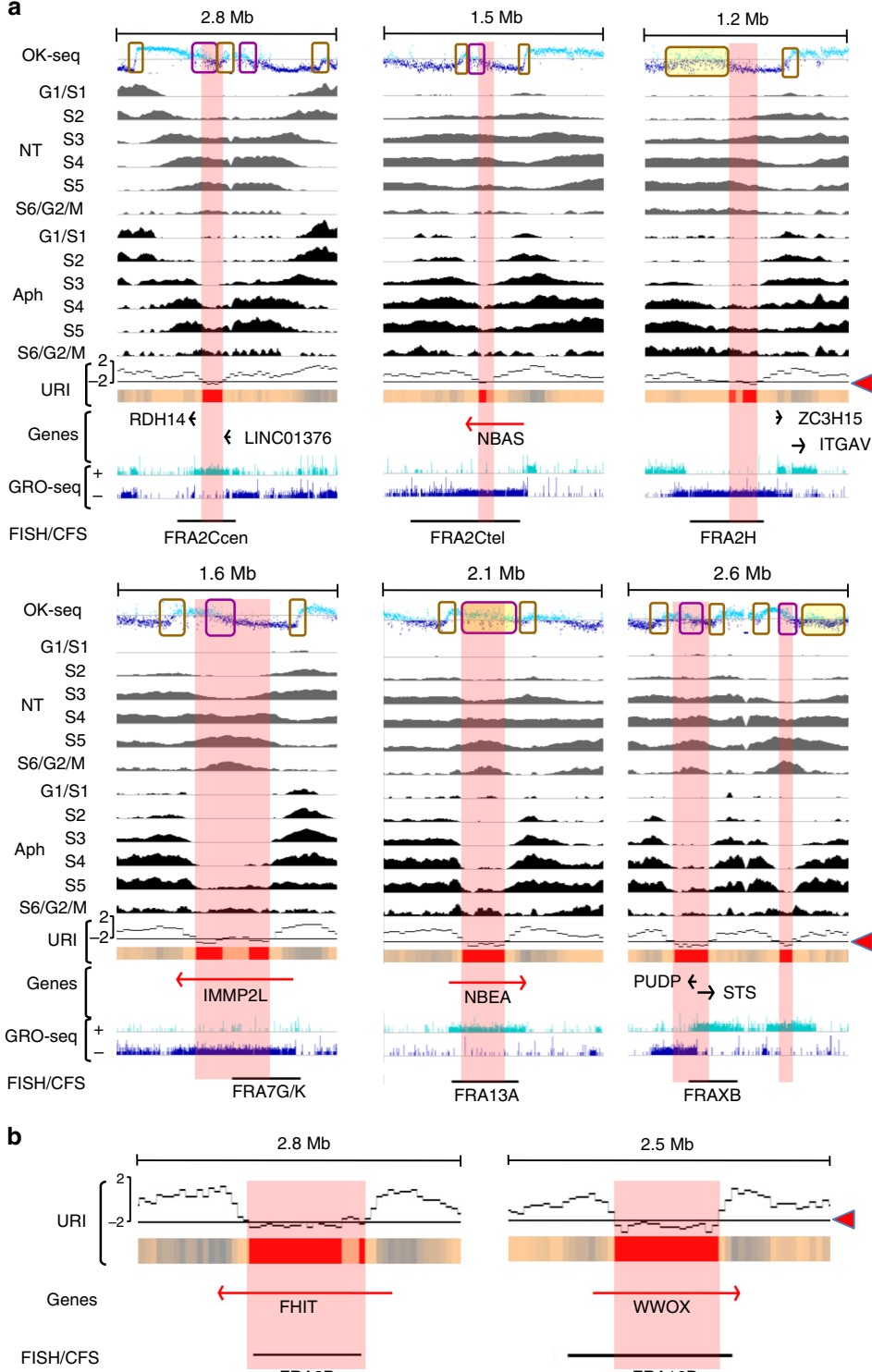

**Fig. 3 T-SDR/SDWs colocalize with CFSs fine-mapped by FISH. a** T-SDRs/SDWs positioning relative to the instable regions. OK-Seq, Repli-Seq and GRO-Seq profiles, and T-SDRs/SDWs (presented as in Fig. 2c) are shown along regions hosting the indicated genes/CFSs. These sites were chosen, because they display a break frequency ≥ 1% in normal lymphocytes[32] and have been fine-mapped by FISH (black bars) in lymphocytes[19]. The T-SDRs/SDWs are included in, or at least partially overlap, the instable regions. The genomic regions displayed are as follows: chr2:17.8-20.6 Mb; chr2:14.7-16.3 Mb; chr2:186.4-187.9 Mb; chr7:109.8-111.5 Mb; chr13:34.9-37 Mb; chrX:6.2-8.9 Mb. **b** T-SDRs positioning relative to the instable regions hosting FHIT/FRA3B and WWOX/FRA16D relative to the URI profiles. The genes, URI profiles, T-SDRs/SDWs and instable regions mapped by FISH are presented as in **a**. Source data are provided as a Source Data file.

results show that the precise position of the initiation zones flanking large genes and their relative efficiency and firing time determine the localization of under-replicated regions upon fork slowing (Fig. 2d).

**The URIs are independent of fork to transcription direction**. In addition, we noticed that all T-SDRs/SDWs are flanked by regions along which the URI decreases progressively and regularly over 150–250 kb, independently of the gene orientation and fork directionality (Fig. 1f left panel, Fig. 2c, e and Supplementary Fig. 2a, b). This decrease is nearly symmetric on both sides of centred T-SDRs/SDWs (Fig. 1f, 2c left panels and 3a). When the T-SDRs/SDWs are asymmetric, the slope of URI decrease may also be asymmetric, but remains progressive and regular on each side of the T-SDR/SDW (Fig. 2c right panel and Supplementary Fig. 2a).

Overall, 93% of the T-SDRs display flanking regions along which the URI decreases over rather similar distances (150–250 kb), independently of the orientation and size of the genes (Fig. 2e, upper panel). In contrast, the size of the T-SDRs does correlate with the size of the large genes (Supplementary Data 1). In addition, the kinetics of URI decrease is nearly similar for all T-SDRs as shown by the weak dispersion of the URI curves (Fig. 2e, lower panel). Noticeably, the dispersion is slightly more important at 3′- than 5′-sides, in agreement with the fact that the upstream initiation zones are most often precisely positioned on the gene promoter, while the downstream ones are less strictly associated with gene 3′-end. In addition, the remarkable symmetry of the URI decrease at the 5′- and 3′-flanking regions of the T-SDRs indicates that the replication delay is independent of the fork direction relative to the transcription direction (lower panels of Fig. 2e). Similar results were obtained for T-SDWs (Supplementary Fig. 2b).

**T-SDRs/SDWs colocalize with CFSs**. To determine whether T-SDRs/SDWs colocalize with CFSs, we mapped CFSs on R-banded chromosomes (Methods) from JEFF cells treated for 16 h with 600 nM Aph. Scoring of 300 metaphase plates yielded a total of 320 breaks in 59 loci (Supplementary Data 3), among which 39 showed a break frequency ≥1% (Supplementary Fig. 3a). Among these 59 loci, 58 co-map with CFSs that have been previously localized by an extensive G-banding analysis of primary lymphocytes from three healthy donors[32] (Supplementary Fig. 3a) and the last one (FRA3O) has been described in fibroblasts and epithelial cells. It is noteworthy that break frequencies at a given site may however vary between the different studies, which is not surprising as they also vary in lymphocytes of different donors[32]. Thank to this good concordance, we could use the large amount of data available from lymphocytes. Notably, the fine mapping of several CFSs by fluorescence in situ hybridization (FISH)[19] allowed us to precisely compare the position of instable regions with that of T-SDRs/SDWs. We found a very good concordance between the position of T-SDRs/SDWs and the most instable region of all eight fine-mapped CFSs with a break frequency ≥ 1% in primary lymphocytes (Fig. 3a, b).

Among 59 chromosome bands displaying breaks in our conventional cytogenetic mapping, 47 (80%) contain T-SDRs and/or T-SDWs (Supplementary Data 3), which confirms the correlation between T-SDRs/SDWs and CFSs. Noticeably, the FHIT and WWOX genes that host, respectively, FRA3B and FRA16D, two major sites in primary lymphocytes and in JEFF lymphoblasts (Supplementary Data 3), display the largest T-SDRs, ~800 kb each (Fig. 2c and 3b). Moreover, 38 T-SDRs (83%) and 32 T-SDWs (71%) nest in cytogenetic bands hosting CFSs mapped in current study and/or in primary lymphocytes[32]

(Supplementary Data 1). T-SDRs/SDWs are therefore a hallmark of CFSs.

Twelve CFSs mapped in JEFF cells by conventional cytogenetics remain free of T-SDRs/SDWs. Among them, FRA1E and FRA6C, each containing a large active gene (DPYD and CDKAL1, respectively), are poor in initiation events and are replicated in the second half of S phase in untreated cells, suggesting that a few T-SDRs/SDWs may have escaped detection (Supplementary Fig. 3b). For example, the minimum URI of CDKAL1 is −1.83, slightly above the −2 cut-off. However, raising the cut-off to 1.8 to include this gene resulted in a high number of false positive regions. Nevertheless, our method remains highly reliable, as we identified 39 out of the 42 CFSs nested in large expressed domains (93%). Because of the large size of cytobands and in the absence of guides such as T-SDRs/SDWs or FISH mapping, the remaining sites cannot be further studied.

**CFS does not rely on transcription–replication encounters**. The properties of T-SDRs/SDWs we described above do not favour the model in which CFS instability results from R-loops formed upon head-on replication–transcription collisions. To directly check this hypothesis, we dissociated transcription–replication encounters from transcription-induced replication initiation clearing. As building of new origins is prevented in the S phase[26], we set up experimental conditions in which metaphase plates observed at the end of the experiment correspond to cells that were already engaged in the S phase when transcription was inhibited.

We used triptolide (Tpl)[33] to inhibit transcription and determined the shorter time of treatment able to clear genes of ongoing transcription (Fig. 4a). We found that 3 h of treatment are sufficient to clear genes of small or moderate size of nascent RNAs (Supplementary Fig. 4a) but longer times are necessary to progressively clear the 1.5 Mb-long FHIT gene (Fig. 4a). Noticeably, the clearing kinetics across FHIT (≈4 kb/min) agrees with the genome-wide RNA polymerase II elongation rates previously measured in the body of genes poor in G/C and in exons[34], two features common to large genes hosting CFSs. We chose to pre-treat the cells with Tpl for 5 h, the shorter time required to clear the FHIT body from most nascent RNAs. We also reduced the time of Aph treatment from 16 h to 7 h, which is sufficient to induce chromosome breaks at CFSs (Fig. 4b left panel), including FRA3B (Fig. 4b right panel). Importantly, compared with cells treated with Aph alone, the mitotic index was not strongly reduced in cells pre-treated for 5 h with Tpl alone, then kept for another 7 h with Tpl and Aph (Tpl + Aph) (Supplementary Fig. 4b). Although treatment with Tpl alone for 12 h impacts the mitotic index, the mitotic flow remains essentially dictated by Aph in cells grown in Tpl + Aph, which makes the two conditions easy to compare.

The percentage of Bromodeoxyuridine (BrdU)-labelled metaphase plates recovered from cell populations pulse labelled at the beginning of the experiment was then determined (Fig. 4c). We found that ~75% of the metaphase plates were labelled when untreated cells were pulsed as indicated. This percentage increases to ~90% in cells treated with Tpl alone, with Aph alone or with Tpl + Aph (Fig. 4c). Quantification of all chromosome breaks in metaphase plates recovered from cell populations grown in each condition showed that Tpl alone, at least up to 12 h of treatment, does not increase the percentage of metaphase breaks relative to untreated cells, while 7 h of Aph treatment increases this percentage by a factor of three. Strikingly, Tpl fails to rescue chromosome breaks elicited by Aph treatment (Fig. 4d and Supplementary Fig. 4c). These results were supported by FISH experiments with FHIT-specific probes showing that Tpl does

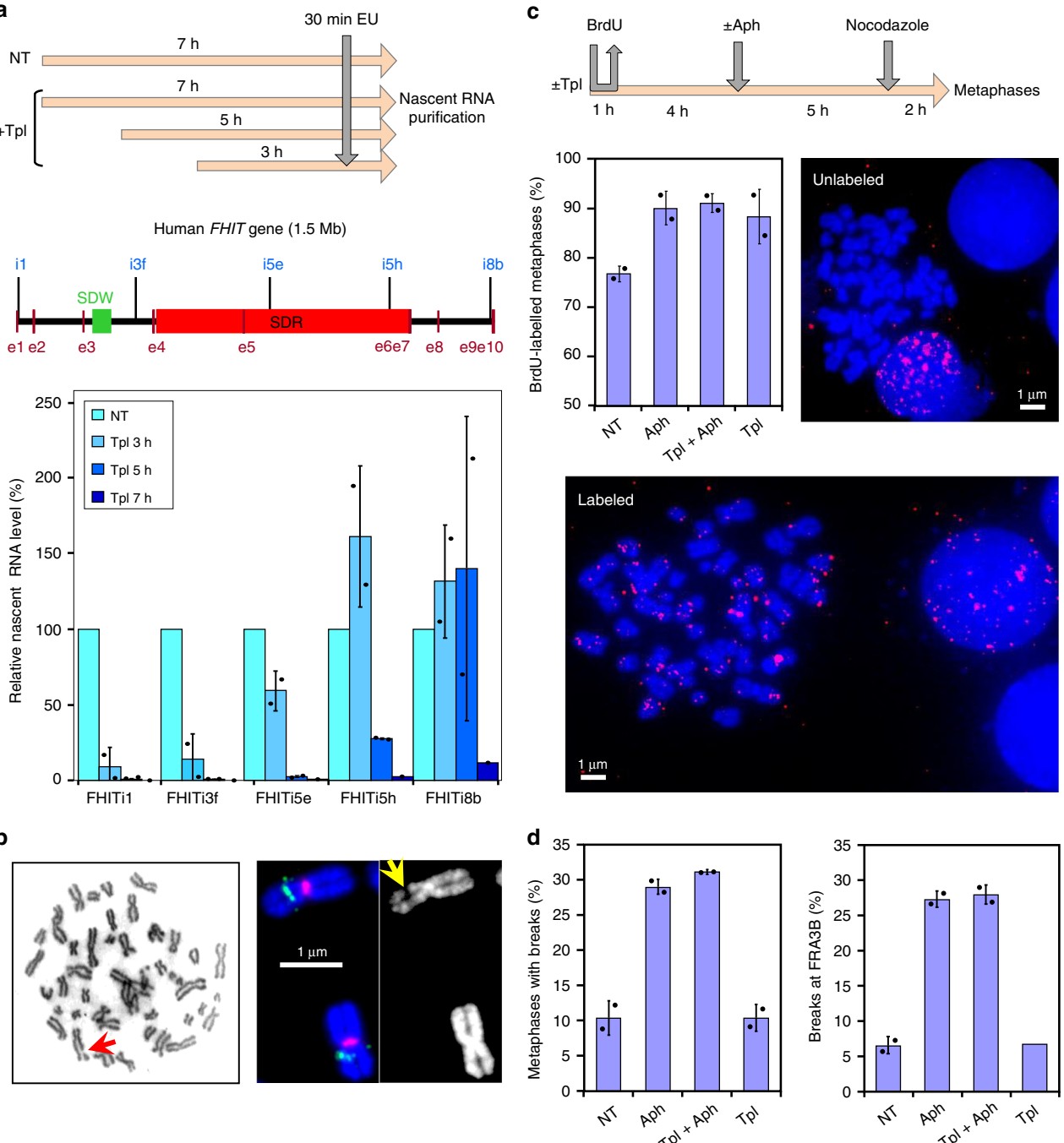

**Fig. 4 Short-term inhibition of transcription by triptolide does not impact CFS instability. a** Determination of the minimal time permitting clearing of the *FHIT* gene from ongoing transcription. Upper panel: scheme of the experiments. Untreated cells (NT) and cells treated with 1 μM triptolide (Tpl) for the indicated periods of time were pulse labelled with EU for 30 min before recovery. Nascent RNAs were prepared by the Click-It method and quantified by RT-qPCR. Middle panel: map of the human *FHIT* gene with the position of exons (e1–e10), T-SDW/SDR and intronic primer pairs (i1–i8b) used for quantification. Lower panel: quantification of nascent RNA along *FHIT* at the different times of Tpl treatment. **b** Examples of chromosome breaks in cells treated with 600 nM Aph for 7 h. Left panel: metaphase plate displaying a break (red arrow) on Giemsa-stained chromosomes. Right panel: break at FRA3B (yellow arrow) visualized after DAPI staining and FISH with probes specific to the *FHIT* gene (green) and to the centromere of chromosome 3 (red). Contrast of DAPI-stained chromosomes (rightmost panel) was enhanced to better show the break. **c** Upper panel: scheme of the experiments: cells were treated with 1 μM triptolide and 600 nM Aph, alone or in combination, for the indicated times before metaphase preparation. In all conditions, cells were pulse labelled for 1 h with 30 μM BrdU at the beginning of the experiments and treated for 2 h with 200 nM nocodazole at the end of experiments to enrich the populations in metaphase cells. Lower panels: the fraction of BrdU-labelled metaphases was scored after DAPI staining and immunofluorescence revelation with anti-BrdU antibodies. Examples of BrdU-labelled and unlabelled metaphases are shown. Note also the presence of labelled and unlabelled interphase nuclei. **d** Determination of the frequencies of total chromosome breaks (left panel) and breaks at FRA3B (right panel). Breaks were scored as in **b**. Experiments shown in a, **c** and **d** were carried out twice and the error bars represent the SD. Source data are provided as a Source Data file.

not suppress Aph-induced breaks at *FHIT*/FRA3B (Fig. 4d and Supplementary Fig. 4c). We conclude that transcription–replication encounters are not responsible for the instability of CFSs in these cells.

## Discussion

Several mechanisms have been proposed to explain CFS instability, most of them postulating that CFS replication occurs in late S phase and is further delayed upon stress. CFSs therefore tend to remain under-replicated up to mitotic onset. However, this hypothesis stems from the study of only three sites[11] and a previous genome-wide analysis has failed to establish a link between CFS instability and replication features[35]. Therefore, it is unclear to what extent under-replication is common and specific to CFSs. Here we searched blind for regions delayed/under-replicated in lymphoblasts treated with Aph and studied the properties of identified loci. The main category of delayed loci corresponds to the so-called T-SDRs/SDWs that nest in large genes or adjacent genes behaving similar to a single large transcribed domain, the body of which replicates in the second half of S phase. These genes display significantly higher transcription levels than the bulk genes with similar replication timing. We then showed that almost all T-SDRs and part of the T-SDWs overlap with the most fragile region of fine-mapped CFSs or nest in cytogenetic bands hosting CFSs in primary lymphocytes for those not mapped by FISH. Noticeably, our method proved to be highly sensitive and efficient for genome-wide mapping of CFSs, as 32 out of 39 (82%) CFSs displaying a break frequency higher than 1% in conventional cytogenetic experiments in JEFF lymphoblasts were identified here at the sequence level, including 24 not yet fine-mapped.

It has been proposed that CFS instability results from fork barriers raised by sequences prone to form secondary structures[18,19,36]. At least four types of observations argue against this model: (i) analysis of the nucleotide sequence of several FISH-mapped CFSs has provided contrasting results regarding the presence of such roadblocks at a substantial proportion of the sites[37]. In addition, recent genome-wide mapping of CFSs has failed to identify sequence features that are specific and common to CFSs[9]. (ii) The fact that CFS setting is tissue-dependent[11] strongly questions this hypothesis, at least in this simple form (see below). (iii) We did not find abrupt drops of URI in the vicinity of specific sequences. On the other hand, global analyses revealed that the URI decreases progressively and regularly across 150–250 kb and shows a near-symmetrical slope on each side of the T-SDRs/SDWs. To account for these results, sequences forming fork barriers should distribute in a very particular manner, similar in all large fragile genes, a feature that has never been reported. For example, stretches of AT-dinucleotide repeats, a type of sequence proposed to create fork barriers, do lie in the *FHIT* and *WWOX* genes[18,19] but the distribution of these sequences does not account for the properties mentioned above. (iv) Molecular combing analyses of fork progressing across the *FHIT* gene in lymphoblasts treated or not with Aph did not detect any site-specific fork slowing or stalling[38]. We conclude that sequence-dependent fork barriers do not account for CFS instability, except in genetic contexts that impair resolution of secondary structures[39–41]. In striking contrast, our data clearly indicate that the slopes of URI decrease, which culminates at T-SDRs/SDWs, correlate with the relative efficiency, firing time and localization of the initiation zones flanking the large fragile genes. The present work focuses on T-SDRs/SDWs, so that our conclusions are specific to CFSs. Poly dA:dT tracks have been involved in the instability of early-replicating fragile sites (ERFS), another type of instable sequences associated with short, highly

transcribed and early-replicating genes[42,43]. In addition, a recent report has shown that a 3.4 kb dA:dT dinucleotide sequence targeted to the 40 kb long early-replicating HGPRT gene triggers instability of this house-keeping gene[44]. Therefore, sequences able to form secondary structures may become instable in some chromosome contexts, notably ERFS, but not in the context of modestly transcribed large late-replicating genes.

Other models rely on the association of CFSs with large expressed genes. One model proposes that CFS instability is due to head-on encounter of the transcription and replication machineries[17,24]. However, as mentioned above, the URI decreases symmetrically across hundreds of kb on each side of the T-SDRs/SDWs, showing that the delay of replication completion is independent of fork direction relative to transcription direction. The finding that R-loops have a short half-life[45] also argues against a major role of these structures in CFS instability. Cells indeed contain many factors, such as FANCD2[46], to prevent harmful R-loop accumulation and subsequent fork stalling. Together, our results strongly suggest that cells can cope with dynamic R-loops that may form in the body of large genes except in particular genetic contexts, such as deficiencies in proteins of the FANC pathway that lead to increased CFS instability[2,47–49] when R-loops abnormally accumulate[50].

To strengthen this conclusion, we directly checked the impact of transcription–replication encounters on CFS instability. To this aim, we treated the cells with Tpl to clear the genes from the transcription machinery and nascent RNA[34]. The time course we chose allowed us to analyse metaphase plates coming from cells in which large genes were cleared while already engaged in the S phase, a period during which the setting of new origins is prevented by redundant pathways[26]. We found that Aph-induced breaks at CFSs were not suppressed under these conditions, again showing that transcription–replication encounters, whether head-on or co-directional, are not a significant source of CFS instability. Our results also strongly argue against the model proposing that CFS instability results from a combination of R-loops and secondary DNA structures[18,19].

The results presented here point to initiation paucity as the major cause of CFS instability. Consistently, MCM7-[51] and ORC2-[52] ChIP-Seq experiments have revealed an under-representation of these components of the pre-replication complex[26] across genes associated with CFSs. Transcription-mediated chase of initiation events has been reported in the bulk genes[20–24] and in large fragile genes[25] of various organisms. Transcription therefore shapes the replication initiation profile along transcribed sequences independently of their size and replication timing. However, fork slowing perturbs more drastically replication completion of loci replicated by long-travelling forks, a feature determined both by the size of the transcribed domain and by the degree of initiation paucity of the gene body, the latter property being controlled by the level of transcription[25].

In contrast to other models, the specific delay of replication completion elicited upon slowing of long-travelling forks together with the relative properties of the initiation zones flanking the large genes readily account for the slope of URI decrease on both sides of the T-SDRs/SDWs. Moreover, the size of the regions along which the URI decreases on each size of the T-SDRs/SDWs (i.e., 150–250 kb) is consistent with the 300 kb threshold repeatedly reported for genes hosting CFSs[8,9,12]. We also showed here that the flanking initiation zones often fire in the first half of S phase, sometimes very early, indicating that long-travelling forks are strongly involved in late replication completion of the large gene bodies in unchallenged cells and in their under-replication upon slowing. Therefore, transcription-dependent modulation of the initiation programme dictates the tissue-dependent landscape of CFSs.

## Methods

**Cell culture.** JEFF cells (human B lymphocytes immortalized with the Epstein–Barr virus) were grown as previously described[38]. Aph and Tpl were obtained from Merck (A0781 and T3652, respectively).

**Repli-Seq experiment.** The technique was essentially as described by Hansen et al.[27]. Exponentially growing lymphoblastoid cells (~$200 \times 10^6$) were pulse-labelled with 50 μM BrdU (Merck, B-5002) prior to recovery. Untreated cells were pulse-labelled for 1 h and Aph-treated cells (600 nM, for a total of 16 h) were labelled during the last 3 h (Fig. 1a). Cells were then fixed in 70% ethanol and incubated overnight at 4 °C in the presence of 15 μg/ml Hoescht 33342 (ThermoFisher Scientific, H3570). Cells were re-suspended in 1× phosphate-buffered saline and sorted in six fractions at a time by flow cytometry based on their DNA content using a BD Biosciences INFLUX cell sorter. The first fraction contains cells in the second half of the G1 phase plus very early S phase (called G1/S1), the rest of S phase was divided into four fractions (S2 to S5) and the last fraction contains cells in very late S, G2 and M phases (S6/G2/M) (Supplementary Fig. 1a). To check the fractionation quality, the post-sorted cells, already stained with Hoescht 33342, were directly re-analysed by flow cytometry. Cells were incubated overnight in lysis buffer (50 mM Tris-HCl pH 8, 10 mM EDTA, 0.1% SDS, 50 μg/mL RNAse A, 100 μg/mL Proteinase K). DNA was purified from the lysates by phenol extraction. Four micrograms of genomic DNA was fragmented to a mean size of 500 bp on a Covaris S220 instrument (peak power: 140, duty factor: 10%, cycle/burst: 200, time: 80 s). Fragmented DNA was treated with the End-Repair module (New England BioLabs, E6050) and the A-Tailing module (New England BioLabs, E6053), according to the manufacturer's recommendations. Illumina Truseq indexed adapters were ligated on the resulting fragments, using the Quick Ligation module (New England BioLabs, E6056), according to the manufacturer's recommendations. Following heat denaturation, BrdU-labelled DNA was isolated by immunoprecipitation using an anti-BrdU monoclonal antibody (BD Biosciences, 347580). Immunoprecipitated fragments were amplified for ten cycles using the KAPA Hifi DNA polymerase (KAPABiosystems, KK2502) and the resulting libraries were purified with AMPure XP beads (Beckman Coulter, A63881). Illumina libraries were pooled and sequenced on a NextSeq 500 instrument on Paired-end $2 \times 43$ or $2 \times 75$ bases, using a NextSeq 500/550 High Output Kit v2.

**Repli-Seq data processing.** The Repli-Seq data were demultiplexed using the distribution of CASAVA software (CASAVA-1.8.2 bcl2fastq2 v2.18.12). Illumina adapters were removed using Cutadapt-1.15, keeping only reads with a minimal length of ten nucleotides. The reads were mapped on the human genome (Hg19), the chicken genome (galGal4, BrdU-labelled DNA as positive control) and the salmon genome (GCF_000233375.1_ICSASG_v2, unlabelled DNA as negative control) using bwa-0.6.2-r126. The mapped data were then processed as previous described[53], with the following modifications. The PCR duplicates were removed with the Picard tools (http://broadinstitute.github.io/picard) and the paired-end reads that were mapped properly to a unique position of the genome were kept for downstream analysis. The sequence reads located within the regions likely resulting from the sequencing hotspots (defined as the 0.5% windows with the highest amount of reads within 200 bp windows) were also removed. The read density ($D_{w,S_i}$) was then computed for each 50 kb non-overlapping windows ($w$) for each sample $S_i$ corresponding to the different S phase fractions (i = 1–6) as well as for the control sample of cells within entire S phase named S0 (density $D_{w,S0}$). The background levels were then estimated as previously described:[53] a background window in an $S_i$ fraction was defined as a window that is not enriched compared to the control window in the adjacent fraction(s) and enriched in the nonadjacent fraction(s). The replication timing, S50, defined as the moment in S phase, on a scale from 0 (Early) to 1 (Late), at which a given sequence has been replicated in 50% of the cells, was computed by linear interpolation of the enrichment values in the six compartments of S phase. When a region was not significantly enriched in all six $S_i$ periods, no S50 value was attributed (~5% of the genome regions, mostly located close to telomeres or centromeres). The S50 values of biological replicates were strongly correlated to each other ($R > 0.95$, $P < 10^{-15}$) (Fig. 1c and Supplementary Fig. 1). The mean S50 values of the biological replicates were therefore used in the downstream analyses. The raw sequencing data and the processing data are available in Gene Expression Omnibus (GEO) with accession number GSE134709. The raw Repli-Seq data of other lymphoblastoid cells (GM06990 and GM12878) were downloaded from the Encode project (EncodeUwRepliSeq [http://genome.ucsc.edu/cgi-bin/hgFileUi?db=hg19&g=wgEncodeUwRepliSeq]) and the S50 were computed.

**SDR identification.** To identify the genomic loci, the replication of which is specifically delayed upon Aph treatment, we computed the difference between the amount of newly replicated DNA measured by Repli-Seq in Aph-treated cells (Aph) and in non-treated cells (NT). A URI was defined as the Z-score computed on the $\Delta$(Aph – NT) by using all 50 kb window along the human genome, where the $\text{Z-score}_i = \frac{(\text{Aph}_i - \text{NT}_i) - \text{mean}(\text{Aph} - \text{NT})}{\sigma(\text{Aph} - \text{NT})}$, measuring the difference between the Aph and NT signals summed over the six periods $\sum_{j=1,6} \left( \text{Aph}_j - \text{NT}_j \right)$, where the mean

and $\sigma$ were computed by all $\sum_{j=1,6} \left( \text{Aph}_j - \text{NT}_j \right)$ along the genome. The windows with a URI < −2 ($n = 330$, 0.54% of genome) were defined as windows with a significantly delay ($P < 0.05$), called SDWs. To limit the false positive results, the windows with too low or too high amounts of reads were removed and only the windows with average coverage, i.e., mean(NT, Aph), between 20 and 40 after normalization, were retained. The SDWs were frequently close to each other and formed clusters (Supplementary Fig. 1). Hence, the close windows ($n \geq 2$) passing the filtering process and separated by a distance < 250 kb (maximum distance between adjacent SDWs located within fine-mapped CFSs) were then merged and defined as an SDR.

**Metadata analyses.** The RNA-Seq data of GM12878 cells from the ENCOE project (GEO: GSM758559, GSM758559_hg19_wgEncodeCshlLongRna-SeqGm12878CellPapGeneGencV7.gtf [https://www.ncbi.nlm.nih.gov/geo/download/?acc=GSM758559&format=file&file=GSM758559%5Fhg19%5FwgEncodeCshlLongRnaSeqGm12878CellPapGeneGencV7%2Egtf%2Egz]) were used. Annotation of genes was retrieved according to gencode V7. The level of transcription was calculated in RPKM (reads per kilobase per mmillion mapped reads) for each protein coding gene. The genes with RPKM > 0.0001 for both biological replicates were kept and the mean values of the two replicates were used. Raw GRO-Seq data of GM12878 cells generated in ref. [29] (GEO: GSM1480326) were used for measuring the transcriptional level of each gene. GRO-seq read densities on the corresponding strand were calculated for all 1 kb non-overlapping windows along each gene (gencode v7) and the median value was then computed for each gene. Inter-origin distances in GM06990 lymphoblastoid cells were calculated by using replication origins identified by Bubble-Seq[30]. Only the Bubble-Seq origins identified in at least two biological replicates were retained in the analysis. The RFD data of GM06990 cells determined by sequencing of Okazaki fragments (OK-Seq)[31] as well as the replication initiation zones, termination zones and regions replicated by unidirectional replication forks were used (SRA: SRP065949 [https://www.ncbi.nlm.nih.gov/sra/?term=SRP065949]). The Repli-Seq, Ok-Seq, Bubble-Seq and GRO-Seq data were analysed using custom scripts written in Python (v2.7.9) and R (v3.4.4), and the data were visualized with the Integrative Genomics Viewer[54].

**Cytogenetic analysis.** The populations were enriched in metaphasic cells by 2 h of treatment with 200 nM nocodazole (Merck, M1404) prior to cell recovery. Total breaks were counted on metaphase plates stained with Giemsa (Prolabo) without pre-treatments to obtain a homogeneous staining of the chromosomes. Preparations were then de-stained in 70% ethanol and treated to reveal R-bands as previously described[55]. Preparations were re-stained with Giemsa and the previously detected breaks were localized relative to the bands. FISH on metaphases, Giemsa counter-staining and immunofluorescence revelation of BrdU-labelled DNA or FHIT-specific FISH probes were carried out as previously described[38,56]. The chromosome 3 centromeric probe was from Aquarius Probes (LPE03R).

**Nascent RNA isolation and quantification.** Five-ethynyl-uridine (EU) (Life Technologies, E10345)) was added to the cell culture medium (1 mM final) during the last 30 min of treatments (Fig. 4a). Total RNA was extracted with the miRNeasy kit (Qiagen, 217004) and nascent RNA was isolated using the Click-It Nascent RNA Capture kit (Invitrogen, 10365) and streptavidin-coated magnetic beads (Dynabeads MyOne Streptavidin T1, Invitrogen 11754). After dissociation of bead-bound RNA by heating (70 °C, 5 min), cDNA synthesis was carried out using the Superscript VILO cDNA Synthesis kit (Invitrogen, 11754). RNA/cDNA hybrids were then incubated for 5 min at 85 °C and quantification was carried out by quantitative PCR with specific primer pairs (Fig. 4 and Supplementary Fig. 4; sequences of primers are available upon request).

**Reporting summary.** Further information on research design is available in the Nature Research Reporting Summary linked to this article.

## Data availability

All sequencing files and processed count matrices were deposited in Gene Expression Omnibus (GEO) under accession number GSE134709. Previously published data (accessions numbers) have been included in the Methods section where appropriate. The source data for Figs. 1b–e, 2a, e and 4a, c, d, and Supplementary Figs. 1b–d, 2b, 3a and 4a, b are provided as a Source Data file. The Integrative Genomics Viewer session for Figs. 1f, 2c and 3a, b and Supplementary Figs. 1e, 2a and 3b is also included in the Source Data file. All data are available from the authors upon reasonable request.

## Code availability

The computer codes and further processing data are available on the GitHub repositories of the team ([https://github.com/CL-CHEN-Lab/RepliSeq] and [https://github.com/CL-CHEN-Lab/CFS-Seq]).

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

## Acknowledgements

The three teams contributing to the work (M.D., C.-L.C., C.T.) have been supported by the Fondation pour la Recherche Médicale (FRM) (programme DBI20131228560). M. D.'s team is also supported by the Institut National du Cancer (INCa) (subvention AAP INCa 2017 -PLBIO17-194). C.-L.C.'s team is also supported by the ATIP-Avenir programme and by the Plan Cancer programme. S.E.-H. was supported by the FRM programme. D.A. was supported by a fellowship from the Ligue Nationale Contre le Cancer. We acknowledge the Imaging and Cytometry Platform (UMS 3655 CNRS/US 23

INSERM) of Gustave Roussy Cancer Campus for assistance with cell sorting, the High-throughput sequencing facility of I2BC for its sequencing and bioinformatics expertise, and Allyson Holmes for her critical reading of the manuscript.

## Author contributions

M.D., C.-L.C. and C.T. conceived the project and analysed the results. M.D. and C.-L.C. wrote the paper. O.B. and S.E.-H. contributed to writing and preparation of figures and tables. C.T. provided critical revision. O.B. contributed to and directed D.A., M.S. and A.-M.L. bench work and analysed the results. S.E.-H. and C.-L.C. designed and performed bioinformatics and statistical analyses. D.A., S.K., M.S., V.N.-K. and A.-M.L. contributed to biological experiments, including cell culture, cell sorting, immunopre-cipitation of BrdU-labelled DNA and molecular cytogenetics. D.A., S.K., M.S., V.N.-K. contributed equally to this work. Y.J. did the Repli-Seq libraries and the sequencing. B.D. did the conventional cytogenetics mapping of CFSs in JEFF cells.

## Competing interests

The authors declare no competing interests.
