## [Peer Review File · Nature Communications]

Reviewers' comments:

Reviewer #1 (Remarks to the Author):

In this manuscript, the authors use genome-wide and single cell techniques to assess how replication timing and transcriptional activity correlate with genome stability. Using Repli-Seq, they define genomic regions where replication timing is strongly perturbed in the presence of aphidicolin (SDWs). Similar to previous reports, they find a strong correlation between late replication, long genes and low transcriptional activity to CFSs. They find that transcriptionally active significantly delayed regions (T-SDR/SDWs) strongly correlate with CFSs. Their novel work shows that CFSs have no correlation with the orientation of replication forks and transcriptional activity—thus, CFSs do not have a strong correlation of head-on vs. co-transcriptional replication orientation. Further, transcriptional activity during S phase is also not necessary for APH-induced replication timing change (RI) or CFS breakage. This manuscript presents novel findings that advance our understanding in fragile site biology. Instead of posing a direct physical hazard, transcription within long, late-replicating genes alters replication timing and origin density. Factors independent of active transcription during S phase drive alterations in replication timing and CFS fragility in cells exposed to aphidicolin.

The manuscript presents novel and exciting data that will be of interest to many researchers in DNA replication and repair. It also furthers our understanding on the underlying causes of CFS instability. In my opinion, it does not require further experimentation or analysis as the scientific experiments are acceptable in their current form. However I recommend a number of editorial changes to the manuscript to clarify certain passages. The manuscript also needs to be edited for grammar, agreement, some sentence structure.

General question: SDR and SDWs are treated similarly throughout most of the manuscript and do not appear to have many differences (except for Figure 1c,e). Do SDRs correlate with CFSs more than “isolated” SDWs? I am unclear if SDRs are distinct from SDWs (figure 1d, 2a, they look the same). Does clustering of SDWs make their effect easier to visualize? There is also a shift in SDWs for 1c as well, which makes sense as this is a visual representation of how they were identified...

Detailed comments

- 1) Sentence line 72-74 – I found this difficult to read, rewrite?
- 2) Sentence 103-104 – difficult to understand with “however, although, not sufficient” all in same sentence.
- 3) Sentence 117-119 – difficult to read
- 4) “Negative (positive) RIs...” this is confusing – if this refers to the difference Delta (aph-NT) then please state.

Would using the terminology replication difference or replication alteration rather than Replication index be appropriate? Replication index is already used to refer to cell proliferation capacity (average # of times a cell has divided), and replicability index refers to the ability to replicate an experiment. I recommend changing the word index.

- 5) Section 153-158 – is this stating that the SDRs exhibit a more profound shift than the SDWs? I think this point gets lost a bit in the S50 numbers...

- 6) Line 160 – change “second half of the S phase” to “second half of S phase” – appears multiple places

- 7) 183 – retardation  delay?
- 8) Sentence 200-202 – “forks traveling long (distances?) contribute to determine their replication timing” – what does this mean?
- 9) Sentence 202-204 – 5 phrases, 4 commas. Hard to follow
- 10) Sentence 216-219 – long.
- 11) Sentence 209-223 -I don't understand this sentence, much of it seems circular. The forks replicating the longest sequences are most changed, and these forks travel the longest? None of this part contributes to the main point that fork retardation/delay is independent of fork orientation to transcription—no difference with head-on vs co-transcriptional.
- 12) “T-SDRs/SDWs co-map with CFSs”  T-SDRs/SDWs overlap/correlate with CFSs
- 13) Line 245-6 - What's a normal lymphocyte? A primary lymphocyte or a “WT” cell line? Please state.
- 14) line 252-3: “which generalizes the notion” – vague; “that T-SDRs/SDWs mark CFSs”; mark  characterize? Rephrase.
- 15) T-SDRs/SDWs are therefore the hallmark of a major category of CFSs.”  “...are a hallmark of CFS”
- 16) 283-288 – I found this intro a bit difficult, mainly: “we analyzed JEFF...cell cycle”. You insert the difficulties of the experiment in the intro, making it a challenge to get to the Q you are trying to answer.
- 17) “although consensual” – not sure this is the phrase you want to use.
- 18) Sentence 325-328 – confusing
- 19) line 333 - “which body” in reference to gene body - use whose? *multiple places
- 20) line 343 “has provided contrasted”  provided contrasting
- 21) Sentence 366-369 – the second half of your sentence seems counterintuitive to the first half. Loss of Fancd2—a protein promoting R loop removal—increases CFS instability. This would argue R loops are involved... The way this sentence/section is written, I think it undermines your point that R loops are unlikely to be the cause of fragility in WT cells... Maybe R loops should be discussed after the Tpl expt? This data also supports your conclusion that R loops as the products of transcription with short half-lives are unlikely to play a major role in CFS fragility in WT cells...
- 22) line 376 - choose  chose
- 23) line 381- “head-to-tail”  co-transcriptional

Reviewer #2 (Remarks to the Author):

Common fragile sites (CFSs) are a major contributor to genome instability and oncogenesis. Even though they have been studied extensively, many questions remain regarding underlying genomic

features that lead to fragility. This manuscript describes the development of a novel, unbiased, sequencing-based method for genome-wide mapping of CFSs based on delay in replication timing. The authors identify 57 so-called delayed regions and show convincing evidence that these are indeed CFSs. They show that CFS fragility is likely caused transcription clearing large stretches of the genome of replication origins, which results in these regions becoming (very) late replicating. They also show evidence that fragility is indeed caused by lack of origin firing within these regions, and not by replication:transcription collisions, as has been proposed by some. Overall, the data presented here agrees with and expands on previously published and proposed models for CFS formation. Furthermore, we believe the methodology described here could easily be adopted by many in the field as a new technique for unbiased CFS identification and mapping in any cell type of interest.

However, the authors also conclude from their data that replication fork stalling at AT-dinucleotide repeats does not contribute to CFS fragility, contrary to several published studies which they do not discuss in this manuscript (eg. PMID: 27768874). We do not believe it's possible to draw such strong conclusions based on the data presented in the current MS, and outline our arguments below. With this in mind, we strongly feel that these sections should be removed from the manuscript before it is suitable for publication.

In summary, we highly recommend that this excellent paper is published without delay.

Specific comments:

As the authors note in their introduction, it has been reported that FANCD2 marks locations of CFSs upon mitotic entry under conditions of replication stress. In the results section, the authors discuss that depending on which cut-off is used to identify SDRs/SDWs, several known CFSs escape detection, or many false positive regions are identified.

Can the authors determine how well FANCD2 ChIP-seq overlaps with the SDWs and SDRs identified here? Can they comment on whether it be useful to combine Repli-seq with FANCD2 ChIP-seq to identify CFSs with higher specificity?

The authors have previously measured fork speed in JEFF cells treated with APH. They then showed that when cells are exposed to 0.6 μ M APH (the dose used here), mean fork speed drops from 1.85Kb/s to 0.2Kb/s. As the authors note in Figures 2c and 3a, SDRs are megabase sized regions of the genome devoid of replication origins. Considering the highly reduced fork speed in the APH-treated cells, one would not expect converging forks from flanking origins to be able to replicate these whole regions before cells enter mitosis. These previous results should be discussed in the context of the data presented here.

It has previously been shown that cells treated with APH perform mitotic DNA synthesis (MIDAS) specifically at CFSs. As shown in Figure S1a, authors sorted cells in a population they dubbed 'S6/G2/M'. Is the BrdU signal detected within SDRs in this population due to 'normal' DNA replication or MIDAS.

In Figure 3b and associated paragraphs in the results and discussion section, the authors claim that replication fork barriers formed by AT-dinucleotide repeats are not responsible for drops in RI, and therefore conclude that these sequences do not contribute to CFS fragility. While we agree that replication timing and transcription are likely the major factors causing CFS fragility, we do not feel AT-repeat fragility can be excluded as a contributing factor. Several arguments can be made against the authors' conclusions:

1) The authors restrict their analysis to 8 AT-repeats found in or near FHIT/FRA3B and WWOX/FRA16D. As indicated by the authors, 4/8 of these AT-repeats lie outside of SDRs. These repeats in fact lie within regions of high RI and when comparing the indicated locations of these repeats in Figure 3b to sites of initiation zones shown in Figure 2c, it becomes clear that these 4

sites either lie within or on the opposite site of initiation zones flanking SDRs. As such, they lie within early replicating, origin-rich regions of the genome where replication fork stalling can easily be rescued by firing a dormant origin.

2) The authors also claim that none of the AT-repeats (again, only 8 examples in two genomic regions are shown) are associated with strong drops in RI scores. However, this analysis is entirely based on binning data within 50Kb windows, and therefore the analysis is not sensitive enough to make any statements regarding specific sequence motifs.

3) Identification of SDRs/SDWs in this study relies on cells treated with a relatively high dose of aphidicolin. As discussed above, the dose of APH used here leads to a drastic decrease in fork speed. As shown in Figure 3b, the 4 AT-repeats in FRA3B and FRA16D are positioned well within the under-replicated regions. Given the very low fork speeds in APH-treated cells, it seems unlikely that forks would reach these AT-repeats before the cells enter mitosis. Based on this data, the authors cannot conclude that fork stalling at AT-repeats does not play a role in cells under conditions of lower replication stress.

4) While the authors rightly point out that AT-dinucleotide repeats are not a common or specific feature of CFSs, several studies have shown that AT-repeats do induce replication fork stalling. One simple model holds that replication fork stalling at AT-repeats is especially detrimental in large regions of the genome devoid of (dormant) origins. In such a situation, a replication fork travelling from a distal origin would need to replicate a much larger region of the genome, increasing the probability that cells enter mitosis before the entire CFS region has been replicated. We do not believe the authors can exclude this model based on the data presented here.

5) The authors also point out that CFSs are tissue dependent, and fragility appears to be largely dependent on transcription of the large genes/gene clusters within the regions. In the absence of transcription, replication origins will not be cleared from these genes and dormant origins will be available to rescue fork stalling at AT-repeats. This fits nicely within the model described above, while still allowing for forks to stall at AT-repeats.

Considering these arguments, we believe it would be better for this manuscript as a whole to remove the sections related to AT-repeats and leave the remaining conclusions to stand for themselves.

Minor remarks:

In Figure 2e, bottom left panel, it is unclear why the box indicating the left-hand initiation zone is drawn starting at G1/S1 for Aph treated cells.

Wherever RI scores are shown in figures, scale should be indicated so readers can assess RI scores throughout the regions.

The author/editors should be make an effort to simplify the text to make it easier to follow.

Reviewers' comments and our point-by-point responses:

Reviewers' comments:

Reviewer #1 (Remarks to the Author):

In this manuscript, the authors use genome-wide and single cell techniques to assess how replication timing and transcriptional activity correlate with genome stability. Using Repli-Seq, they define genomic regions where replication timing is strongly perturbed in the presence of aphidicolin (SDWs). Similar to previous reports, they find a strong correlation between late replication, long genes and low transcriptional activity to CFSs. They find that transcriptionally active significantly delayed regions (T-SDR/SDWs) strongly correlate with CFSs. Their novel work shows that CFSs have no correlation with the orientation of replication forks and transcriptional activity—thus, CFSs do not have a strong correlation of head-on vs. co-transcriptional replication orientation. Further, transcriptional activity during S phase is also not necessary for APH-induced replication timing change (RI) or CFS breakage. This manuscript presents novel findings that advance our understanding in fragile site biology. Instead of posing a direct physical hazard, transcription within long, late-replicating genes alters replication timing and origin density. Factors independent of active transcription during S phase drive alterations in replication timing and CFS fragility in cells exposed to aphidicolin.

The manuscript presents novel and exciting data that will be of interest to many researchers in DNA replication and repair. It also furthers our understanding on the underlying causes of CFS instability. In my opinion, it does not require further experimentation or analysis as the scientific experiments are acceptable in their current form.

Our response: We thank the reviewer for acknowledging the novelty and the quality of our work. We hope that our novel and exciting data will be published soon, as they will be of interest to many researchers in the field.

However I recommend a number of editorial changes to the manuscript to clarify certain passages. The manuscript also needs to be edited for grammar, agreement, some sentence structure.

Our response: The revised manuscript has now been carefully proofread and polished by a native English-speaking colleague and an English-editing professional service.

General question: SDR and SDWs are treated similarly throughout most of the manuscript and do not appear to have many differences (except for Figure 1c,e). Do SDRs correlate with CFSs more than “isolated” SDWs? I am unclear if SDRs are distinct from SDWs (figure 1d, 2a, they look the same). Does clustering of SDWs make their effect easier to visualize? There is also a shift in SDWs for 1c as well, which makes sense as this is a visual representation of how they were identified...

Our response: Thanks for pointing this out. To make it clearer, the following sentences have been added in revised manuscript “The proportion of isolated SDWs not associated with large genes is therefore significantly higher than that of SDRs, suggesting that SDWs mark heterogeneous categories of delayed sequences. We focused below on the 45 T-SDWs, among which 30 display at least one nearby window with an URI close to -2.”.

Detailed comments

1) Sentence line 72-74 – I found this difficult to read, rewrite?

Our response: The sentence has been rewritten.

2) Sentence 103-104 – difficult to understand with “however, although, not sufficient” all in same sentence.

Our response: The sentence has been rewritten.

3) Sentence 117-119 – difficult to read

Our response: The sentence has been rewritten.

4) “Negative (positive) RIs...” this is confusing – if this refers to the difference Delta (aph-NT) then please state.

Our response: The sentence has been rewritten.

Would using the terminology replication difference or replication alteration rather than Replication index be appropriate? Replication index is already used to refer to cell proliferation capacity (average # of times a cell has divided), and replicability index refers to the ability to replicate an experiment. I recommend changing the word index.

Our response: In the revised manuscript, we changed the terminology into Under-Replication Index (URI).

5) Section 153-158 – is this stating that the SDRs exhibit a more profound shift than the SDWs? I think this point gets lost a bit in the S50 numbers...

Our response: Yes. And the sentence has been modified in the revised manuscript as follows “By contrast, 56 out of 57 SDRs (98%) nest in mid-late domains in untreated cells (Fig. 1c, Table S1). Upon stress, the SDRs however present S50 values resembling those of late replicating domains (Fig. 1c), showing that they are much more delayed than all other mid-late regions.”.

Also, the numbers of SDWs in mid-late and late replicating domains have been added in the following sentence “The timing profile of untreated cells shows that isolated SDWs are

equally distributed between mid-late ($0.5 \leq S50 \leq 0.75$, $n=62$) and late ($0.75 \leq S50 \leq 1$, $n=40$) replicating domains and are further delayed upon stress (Fig. 1c, Table S2).”.

6) Line 160 – change “second half of the S phase” to “second half of S phase” – appears multiple places

Our response: Done.

7) 183 – retardation  delay?

Our response: Retardation has been changed to delay.

8) Sentence 200-202 – “forks traveling long (distances?) contribute to determine their replication timing” – what does this mean?

Our response: To make the results clearer and easy to read, this section has been significantly revised. And the indicated sentence has been replaced by the following sentence “Together, our results show that the precise position of the initiation zones flanking large genes and their relative efficiency and firing time determine the localisation of under-replicated regions upon fork slowing (Fig. 2d).”.

9) Sentence 202-204 – 5 phrases, 4 commas. Hard to follow

Our response: The sentence has been rewritten.

10) Sentence 216-219 – long.

Our response: The sentence has been rewritten.

11) Sentence 209-223 -I don’t understand this sentence, much of it seems circular. The forks replicating the longest sequences are most changed, and these forks travel the longest? None of this part contributes to the main point that fork retardation/delay is independent of fork orientation to transcription—no difference with head-on vs co-transcriptional.

Our response: This section has been simplified and rewritten as follows:

“In addition, we noticed that all T-SDRs/SDWs are flanked by regions along which the URI decreases progressively and regularly over 150-250 kb, independently of the gene orientation and fork directionality (Fig. 1f left panel, Fig 2c, 2e; S2 a, b). This decrease is nearly symmetric on both sides of centred T-SDRs/SDWs (Fig. 1f, Fig. 2c left panels, Fig. 3a). When the T-SDRs/SDWs are asymmetric, the slope of URI decrease may also be asymmetric, but remains progressive and regular on each side of the T-SDR/SDW (Fig. 2c right panel, S2a). Overall, 93% of the T-SDRs display flanking regions along which the URI decreases over rather similar distances (150-250 kb), independently of the orientation and size of the genes (Fig. 2e, upper panel). In contrast, the size of the T-SDRs does correlate with the size of the large genes (Table S1). In addition, the kinetics of URI decrease is nearly similar for all T-

SDRs as shown by the weak dispersion of the URI curves (Fig. 2e, lower panel). Noticeably, the dispersion is slightly more important at 3' than 5' sides, in agreement with the fact that the upstream initiation zones are most often precisely positioned on the gene promoter while the downstream ones are less strictly associated with gene 3' end. In addition, the remarkable symmetry of the URI decrease at the 5' and 3' flanking regions of the T-SDRs indicates that the replication delay is independent of the fork direction relative to the transcription direction (lower panels of Fig. 2e). Similar results were obtained for T-SDWs (Fig. S2b)''

12) “T-SDRs/SDWs co-map with CFSs”  T-SDRs/SDWs overlap/correlate with CFSs

Our response: “co-map” has been changed to “co-localized”.

13) Line 245-6 - What’s a normal lymphocyte? A primary lymphocyte or a “WT” cell line? Please state.

Our response: “normal” lymphocyte has been changed to “primary” lymphocyte.

14) line 252-3: “which generalizes the notion” – vague; “that T-SDRs/SDWs mark CFSs”; mark  characterize? Rephrase.

Our response: The sentence has been revised as follows: “Amongst 59 chromosome bands displaying breaks in our conventional cytogenetic mapping, 47 (80%) contain T-SDRs and/or T-SDWs (Table S3), which confirms the correlation between T-SDRs/SDWs and CFSs.”.

15) T-SDRs/SDWs are therefore the hallmark of a major category of CFSs.”  “...are a hallmark of CFS”

Our response: The sentence has been revised as “T-SDRs/SDWs are therefore a hallmark of CFSs”.

16) 283-288 – I found this intro a bit difficult, mainly: “we analyzed JEFF...cell cycle”. You insert the difficulties of the experiment in the intro, making it a challenge to get to the Q you are trying to answer.

Our response: This section has been revised “The properties of T-SDRs/SDWs we described above do not favour the model in which CFS instability results from R-loops formed upon head-on replication-transcription collisions. In order to directly check this hypothesis, we dissociated transcription-replication encounters from replication initiation clearing. Since building of new origins is prevented in the S phase²⁶, we set up experimental conditions in which metaphase plates observed at the end of the experiment correspond to cells that were already engaged in the S phase when transcription was inhibited.”

17) “although consensual” – not sure this is the phrase you want to use.

Our response: “although consensual” has been replaced with “However”

18) Sentence 325-328 – confusing

Our response: The sentence has been rewritten as follows: “and a previous genome-wide analysis has failed to establish a link between CFS instability and replication features³⁵”.

19) line 333 - “which body” in reference to gene body - use whose? *multiple places

Our response: This has been corrected.

20) line 343 “has provided contrasted”  provided contrasting

Our response: Done.

21) Sentence 366-369 – the second half of your sentence seems counterintuitive to the first half. Loss of Fancd2—a protein promoting R loop removal—increases CFS instability. This would argue R loops are involved... The way this sentence/section is written, I think it undermines your point that R loops are unlikely to be the cause of fragility in WT cells... Maybe R loops should be discussed after the Tpl expt? This data also supports your conclusion that R loops as the products of transcription with short half-lives are unlikely to play a major role in CFS fragility in WT cells...

Our response: Thanks for pointing this out. This section has been revised as follows: “The finding that R-loops have a short half-life⁴⁵ also argues against a major role of these structures in CFS instability. Cells indeed contain many factors, such as FANCD2⁴⁶, to prevent harmful R-loop accumulation and subsequent fork stalling. Together, our results strongly suggest that cells can cope with dynamic R-loops that may form in the body of large genes except in particular genetic contexts, such as deficiencies in proteins of the FANC pathway that lead to increased CFS instability^{2, 47-49} when R-loops abnormally accumulate⁵⁰”.

22) line 376 - choose  chose

Our response: Done.

23) line 381- “head-to-tail”  co-transcriptional

Our response: “head-to-tail” has been changed to “co-directional”.

We thank again the reviewer for all his/her comments that helped to improve our manuscript.

Reviewer #2 (Remarks to the Author):

Common fragile sites (CFSs) are a major contributor to genome instability and oncogenesis. Even though they have been studied extensively, many questions remain regarding underlying genomic features that lead to fragility. This manuscript describes the development of a novel, unbiased, sequencing-based method for genome-wide mapping of CFSs based on delay in replication timing. The authors identify 57 so-called delayed regions and show convincing evidence that these are indeed CFSs. They show that CFS fragility is likely caused transcription clearing large stretches of the genome of replication origins, which results in these regions becoming (very) late replicating. They also show evidence that fragility is indeed caused by lack of origin firing within these regions, and not by replication:transcription collisions, as has been proposed by some. Overall, the data presented here agrees with and expands on previously published and proposed models for CFS formation. Furthermore, we believe the methodology described here could easily be adopted by many in the field as a new technique for unbiased CFS identification and mapping in any cell type of interest.

Our response: We thank the reviewer for acknowledging the value and the quality of our work. We hope that the new methodology and novel data described in our study will be published soon, as they will be of interest to many researchers in the field.

However, the authors also conclude from their data that replication fork stalling at AT-dinucleotide repeats does not contribute to CFS fragility, contrary to several published studies which they do not discuss in this manuscript (eg. PMID: 27768874). We do not believe it's possible to draw such strong conclusions based on the data presented in the current MS, and outline our arguments below. With this in mind, we strongly feel that these sections should be removed from the manuscript before it is suitable for publication.

Our response: The section on AT-dinucleotide repeats has been removed from the Results section as suggested by the reviewer. Also, additional discussion and comparisons with previous published studies have been added into the Discussion section, as suggested by the reviewer.

In summary, we highly recommend that this excellent paper is published without delay.

Our response: We thank again the reviewer for his/her strong support for publishing our work.

Specific comments:

As the authors note in their introduction, it has been reported that FANCD2 marks locations of CFSs upon mitotic entry under conditions of replication stress. In the results section, the authors discuss that depending on which cut-off is used to identify SDRs/SDWs, several known CFSs escape detection, or many false positive regions are identified. Can the authors determine how well FANCD2 ChIP-seq overlaps with the SDWs and SDRs identified here?

Can they comment on whether it be useful to combine Repli-seq with FANCD2 ChIP-seq to identify CFSs with higher specificity?

Our response: We agree with the reviewer that it might be interesting to determine how well the FANCD2 ChIP-Seq data overlap with the SDRs/SDWs. However, the FANCD2 ChIP-Seq has been performed only in human osteosarcoma U2OS cells and chicken DT40 cells. Therefore, due to the tissue-dependent instability of CFSs, we cannot compare the data obtained with the two techniques.

The authors have previously measured fork speed in JEFF cells treated with APH. They then showed that when cells are exposed to 0.6 μ M APH (the dose used here), mean fork speed drops from 1.85Kb/s to 0.2Kb/s. As the authors note in Figures 2c and 3a, SDRs are megabase sized regions of the genome devoid of replication origins. Considering the highly reduced fork speed in the APH-treated cells, one would not expect converging forks from flanking origins to be able to replicate these whole regions before cells enter mitosis. These previous results should be discussed in the context of the data presented here.

Our response: These results have been added into the revised manuscript with the following sentences “A previous analysis of unstressed cells by molecular combing has shown that the FHIT gene displays an initiation poor core extending for about 800 kb, and that replication forks travel along the gene at \approx 1.8 kb/min, like in the bulk genome¹¹. In these conditions, convergent forks would need 8-9 hours to complete FHIT replication, in agreement with the replication kinetics observed here (NT in Fig. 2c). Therefore, in addition to the firing time of the initiation zones flanking this large gene, the distance that convergent forks must travel before merging strongly contributes to set the replication timing of the gene body in untreated cells. We found here that this feature is common to large expressed genes (NT in Fig. 1f, Fig. 2c, Fig. 3a). Often, replication could not be completed when fork speed is reduced upon treatment with Aph (Aph in Fig. 1f, Fig. 2c, Fig. 3a), which gives rise to the T-SDRs/SDWs. The distance separating the initiation zones flanking the genes is therefore a major parameter for T-SDRs/SDWs setting.”.

It has previously been shown that cells treated with APH perform mitotic DNA synthesis (MIDAS) specifically at CFSs. As shown in Figure S1a, authors sorted cells in a population they dubbed ‘S6/G2/M’. Is the BrdU signal detected within SDRs in this population due to ‘normal’ DNA replication or MIDAS.

Our response: As mentioned by the reviewer, in the S6/G2/M sample, the BrdU signal detected within SDRs in cells in very late S, G2 and M phases, may be due to either ‘normal’ DNA replication or MIDAS.

In Figure 3b and associated paragraphs in the results and discussion section, the authors claim that replication fork barriers formed by AT-dinucleotide repeats are not responsible for drops in RI, and therefore conclude that these sequences do not contribute to CFS fragility. While we agree that replication timing and transcription are likely the major factors causing CFS fragility, we do not feel AT-repeat fragility can be excluded as a contributing factor. Several arguments can be made against the authors’ conclusions:

1) The authors restrict their analysis to 8 AT-repeats found in or near FHIT/FRA3B and WWOX/FRA16D. As indicated by the authors, 4/8 of these AT-repeats lie outside of SDRs. These repeats in fact lie within regions of high RI and when comparing the indicated locations of these repeats in Figure 3b to sites of initiation zones shown in Figure 2c, it becomes clear that these 4 sites either lie within or on the opposite site of initiation zones flanking SDRs. As such, they lie within early replicating, origin-rich regions of the genome where replication fork stalling can easily be rescued by firing a dormant origin.

2) The authors also claim that none of the AT-repeats (again, only 8 examples in two genomic regions are shown) are associated with strong drops in RI scores. However, this analysis is entirely based on binning data within 50Kb windows, and therefore the analysis is not sensitive enough to make any statements regarding specific sequence motifs.

3) Identification of SDRs/SDWs in this study relies on cells treated with a relatively high dose of aphidicolin. As discussed above, the dose of APH used here leads to a drastic decrease in fork speed. As shown in Figure 3b, the 4 AT-repeats in FRA3B and FRA16D are positioned well within the under-replicated regions. Given the very low fork speeds in APH-treated cells, it seems unlikely that forks would reach these AT-repeats before the cells enter mitosis. Based on this data, the authors cannot conclude that fork stalling at AT-repeats does not play a role in cells under conditions of lower replication stress.

4) While the authors rightly point out that AT-dinucleotide repeats are not a common or specific feature of CFSs, several studies have shown that AT-repeats do induce replication fork stalling. One simple model holds that replication fork stalling at AT-repeats is especially detrimental in large regions of the genome devoid of (dormant) origins. In such a situation, a replication fork travelling from a distal origin would need to replicate a much larger region of the genome, increasing the probability that cells enter mitosis before the entire CFS region has been replicated. We do not believe the authors can exclude this model based on the data presented here.

5) The authors also point out that CFSs are tissue dependent, and fragility appears to be largely dependent on transcription of the large genes/gene clusters within the regions. In the absence of transcription, replication origins will not be cleared from these genes and dormant origins will be available to rescue fork stalling at AT-repeats. This fits nicely within the model described above, while still allowing for forks to stall at AT-repeats. Considering these arguments, we believe it would be better for this manuscript as a whole to remove the sections related to AT-repeats and leave the remaining conclusions to stand for themselves.

Our response: The section on AT-dinucleotide repeats has been removed from the Results section as suggested by the reviewer. Also, additional discussion and comparisons with previous published studies on the role of AT-dinucleotide repeats in genome instability, particularly, the instability of early fragile sites, have been added in the Discussion section: "In striking contrast, our data clearly indicate that the slopes of URI decrease that culminates at T-SDRs/SDWs correlate with the relative efficiency, firing time and localization of the initiation zones flanking the large fragile genes. The present work focuses on T-SDRs/SDWs, so that our conclusions are specific to CFSs. Poly dA:dT tracks have been involved in the instability of early replicating fragile sites (ERFS), another type of unstable sequences associated with short, highly transcribed and early replicating genes^{42,43}. In

addition, a recent report has shown that a 3.4 kb dA:dT dinucleotide sequence targeted to the 40 kb long early replicating HGPRT gene triggers instability of this house keeping gene ⁴⁴. Therefore, sequences able to form secondary structures may become instable in some chromosome contexts, notably ERF5.”

Minor remarks:

In Figure 2e, bottom left panel, it is unclear why the box indicating the left-hand initiation zone is drawn starting at G1/S1 for Aph treated cells.

Our response: The position of the box has been corrected.

Wherever RI scores are shown in figures, scale should be indicated so readers can assess RI scores throughout the regions.

Our response: The scale of RI (called URI in the revised manuscript) scores has been added in the corresponding figures.

The author/editors should be make an effort to simplify the text to make it easier to follow.

Our response: The revised manuscript has now been carefully proofread and polished by a native English-speaking colleague and an English-editing professional service.

We thank again the reviewer for all his/her comments that helped to improve our manuscript.

REVIEWERS' COMMENTS:

Reviewer #2 (Remarks to the Author):

The authors have addressed the concerns of both reviewers, and the paper should be published.